# Characterization of the Interface between Aluminum and Iron in Co-Extruded Semi-Finished Products

**DOI:** 10.3390/ma15051692

**Published:** 2022-02-24

**Authors:** Susanne Elisabeth Thürer, Kai Peters, Torsten Heidenblut, Norman Heimes, Julius Peddinghaus, Florian Nürnberger, Bernd-Arno Behrens, Hans Jürgen Maier, Christian Klose

**Affiliations:** 1Institut für Werkstoffkunde (Materials Science), An der Universität 2, 30823 Garbsen, Germany; kai.peters@stud.uni-hannover.de (K.P.); heidenblut@iw.uni-hannover.de (T.H.); nuernberger@iw.uni-hannover.de (F.N.); maier@iw.uni-hannover.de (H.J.M.); klose@iw.uni-hannover.de (C.K.); 2Institut für Umformtechnik und Umformmaschinen (Forming Technology and Machines), An der Universität 2, 30823 Garbsen, Germany; heimes@ifum.uni-hannover.de (N.H.); peddinghaus@ifum.uni-hannover.de (J.P.); behrens@ifum.uni-hannover.de (B.-A.B.)

**Keywords:** intermetallic phase seam, co-extrusion, aluminum–steel, mechanical properties

## Abstract

Within the framework of the Collaborative Research Center 1153, we investigated novel process chains for the production of bulk components with different metals as joining partners. In the present study, the co-extrusion of coaxially reinforced hollow profiles was employed to manufacture semi-finished products for a subsequent die-forging process, which was then used for the manufacture of hybrid bearing bushings. The hybrid hollow profiles, made of the aluminum alloy EN AW-6082 paired with either the case-hardening steel 20MnCr5, the stainless steel X5CrNi18-10, or the rolling bearing steel 100Cr6, were produced by Lateral Angular Co-Extrusion. Push-out tests on hybrid hollow sections over the entire sample cross-section showed shear strengths of 44 MPa ± 8 MPa (100Cr6) up to 63 MPa ± 5 MPa (X5CrNi18-10). In particular, the influence of force and form closure on the joint zone could be determined using specimen segments tested in shear compression. Locally, shear strengths of up to 131 MPa (X5CrNi18-10) were demonstrated in the shear compression test. From these samples, lamellae for microstructural analysis were prepared with a Focused Ion Beam. Detailed analyses showed that for all material combinations, a material bond in the form of an ultra-thin intermetallic phase seam with a thickness of up to 50 nm could be established.

## 1. Introduction

Over recent decades, efforts have been made to realize lightweight construction concepts by combining different materials. The material combination of aluminum and steel, the most commonly used metallic construction materials, is notably promising for advanced lightweight construction applications [1]. In order to meet local component requirements, various process chains for the manufacture of hybrid components for bulk metal forming are being researched as part of the Collaborative Research Center (CRC) 1153’s *Tailored Forming*. These include the production of hybrid bearing bushings with wear-resistant steel on the bearing surfaces and a significantly reduced mass through the use of aluminum for the rest of the component. In the first manufacturing step, the different materials for the hybrid bearing bushing are joined to semi-finished products by means of co-extrusion. In the second step, these semis are forged into bearing bushings, and thus the joint zone established through co-extrusion undergoes a thermo-mechanical treatment. The process chain developed has been described in more detail in a previous publication [2].

The joining of dissimilar joints by welding has been known for a long time and is currently being investigated, for example, for the combination of steel 316 L with nickel-based Inconel 718/SS for the additive manufacturing of functionally graded materials [3]. Bach et al. have already carried out research on the production of formable hybrid structures for the combination of aluminum and steel using laser-beam welding, electron-beam welding and arc-welding processes, as well as friction-stir welding for use in the thin sheet metal sector [4]. The failure of such material compounds often occurs in areas in which a metallurgical bond is present [5]. This phenomenon of failure between the joining partners aluminum and steel was also found in the case of die forging by Napierala et al. [6]. Such a material bond between the two metals is considered problematic due to the differences in their metallurgical and physical properties. For example, large differences between the individual coefficients of thermal expansion can lead to internal stresses in the joined component [7]. From a metallurgical point of view, the solubility of iron in aluminum is only 0.03 at. % at the lowest solidification temperature of 655 °C. This means that brittle intermetallic phases of the stoichiometry Fe_2_Al_7_, FeAl_3_, Fe_2_Al_5_ and FeAl_2_ are likely to occur [8]. Furthermore, the formation of the superstructures FeAl and Fe_3_Al is possible by multiplication of the elementary cells, in which the atomic layers are occupied by different atoms [9]. At a processing temperature below the melting temperature of aluminum, the intermetallic phases FeAl_3_ and Fe_2_Al_5_ typically emerge [10]. While the intermetallic phases are formed mainly under short-term heat influence, the rearrangement of the atoms in the crystal lattice, which takes place during the formation of the superstructures, requires a longer period of time [11]. The lower symmetry of the crystal structure in the intermetallic phases implies that dislocation movements are difficult, and thus there is an increase in hardness and a reduction in ductility compared to the starting materials.

The properties of these material compounds can be significantly improved by a metallurgical bond between the joining partners, which can be realized by an intermetallic phase seam that is as narrow as possible. Hence, in addition to force and form closure, the concept of *Tailored Forming* also includes material locking, which is crucial for a more uniform load transfer between the different materials, and thus the improvement of the overall mechanical properties of the hybrid components. However, depending on the material combination of the aluminum alloy, as well as the steel grade and the processing parameters, pronounced intermetallic phase seams form frequently along the interface between two metals, and thus the joint zone is considered brittle and will deteriorate the mechanical properties significantly if the phase seam exceeds a width of about 1 µm [12].

In order to avoid that, the intermetallic phase seam reaches a critical width at the end of the process chain, and the parameters for the initial joining process must be tailored in such a way that there is good bonding through the narrowest possible phase seam after the first step of the process chain. In addition, temperature–time profiles during all subsequent process steps have to be taken into account as well. Large layer thicknesses of the intermetallic phase seams generally lead to a deterioration in the properties of material compounds [7]. Since no metal exists that is completely soluble in both aluminum and iron and could therefore contribute to the formation of an interlayer [11], there is always the risk of brittle intermetallic layers emerging during thermo-mechanical processing. Therefore, the mechanisms of the formation of such phases will be presented and ways to control their formation will be discussed in the following. Figure 1a shows that due to extrusion and subsequent die forging the two metals, in this case aluminum and steel, are in intimate contact with each other. Hence, atoms of each of the two metals begin to diffuse rapidly into the opposite phase at sufficiently high temperatures. In this system, the diffusional flux of iron is more pronounced. As a result, local areas of intermetallic phases at the aluminum side of the interface can rapidly be observed [11]. In case of an aluminum–steel compound, FeAl_3_ will be formed initially above 350 °C. At temperatures above 500 °C, characteristic stem-shaped crystals of the phase Fe_2_Al_5_ are formed, which are transformed into FeAl_3_ upon cooling [13]. The local intermetallic phases initially grow along the interface (Figure 1b) until the individual islands finally form larger contiguous areas. Subsequently, the phases grow perpendicular to the joining surface and further types of intermetallic phases emerge. Figure 1c shows a schematic of the resulting microstructure along with the corresponding concentration profiles for aluminum and iron. Note that the second intermetallic phase grows into the steel bulk, while a continuous layer of the initial intermetallic phase extends along the entire joint zone. 

One possible process for joining dissimilar metals with sufficiently narrow intermetallic phases is pressure welding. Press-welded joints generally form sufficiently narrow phase seams so that the tensile strength is not reduced [14]. The tailoring of the microstructure inside the joint zone by means of the subsequent thermo-mechanical treatment, e.g., die forging, will result in better mechanical properties of the finished hybrid component, which is of paramount interest for the development of high-performance hybrid components.

The joint forming of different metals is well established industrially in sheet metal forming. As with tailored blanks, various welding processes can be used to join hybrid massive components. In contrast, profile-shaped hybrid components are manufactured by co-extrusion, which has been established for the production of hybrid semi-finished products consisting of EN AW-6082 and C15, as well as 20MnCr5 steel [2]. These hybrid profiles have been processed further into hybrid bearing bushings. Co-extrusion processes are thus suitable for producing complex-shaped semi-finished products from at least two different materials. Co-extrusion has already been used, for example, to produce Mg-Al macro-composites [15], to press silver-sheathed superconductor materials that were sheathed in copper [16], or to insert wires from spring steel as reinforcement in various profile geometries made of EN AW-6060 [17].

The Lateral Angular Co-Extrusion (LACE) process for the production of hybrid hollow sections from the aluminum alloy EN AW-6082 combined with either C15 or 20MnCr5 steel has previously been described in detail [2,18]. It has been unclear, however, how various steel alloys will affect the properties of the bonding zone since the diffusion paths of individual alloying elements from the joining partners will be different between each other. Therefore, the current study presents the characterization of a promising new combination of compound profiles consisting of EN AW-6082 and the stainless steel X5CrNi18-10 along with the proof of actual material locking in the joint zones for LACE profiles made of different aluminum–steel combinations. The mechanical strength of the joint zone of the produced compound profiles consisting of EN AW-6082 with X5CrNi18-10 and 100Cr6 were investigated by means of push-out tests (POTs) and compared with the reference EN AW-6082/20MnCr5. Furthermore, the influence of force and form closure on the strength of the joint zone was determined by shear compression tests (SCTs) on sample segments. Proof of material-locking is provided by tests on lamellae, prepared using a Focused Ion Beam (FIB).

## 2. Materials and Methods

### 2.1. Co-Extrusion of Aluminum and Steel via LACE

The LACE process has already been described in detail in Refs. [2,18], and thus only the key elements are explained here. Since the LACE process involves feeding a rigid reinforcing element instead of a wire, a die was designed with a mandrel part supported by three support arms in the die housing. Schematically, a section through the tool is shown in front view in Figure 2a. During the LACE process, the aluminum is divided into two material streams by the portholes in the center of the symmetrically designed inlet. Inside the tool, the aluminum is then directed into milled pockets. These pockets were inserted into the die to influence the aluminum flow in such a way that the steel tube is equally clad with aluminum over the entire circumference from the start of the process. After filling the pockets, the aluminum flows around the mandrel part in which the steel tube is guided. Thus, in the LACE process, the reinforcing element is introduced into the die orthogonally to the extrusion direction and is held in place by the clamping cover and the mandrel part. The sectional view shown in Figure 2b illustrates that the LACE extrusion direction is orthogonal to the movement of the ram.

For the LACE experiments, the aluminum alloy EN AW-6082 was used as billet material along with tubes made of different steel grades, i.e., the stainless steel X5NiCr18-10 as well as 100Cr6 as a typical bearing steel, which is particularly relevant for the application. In addition, tubes made of the case-hardening steel 20MnCr5 were used as a reference, to be able to compare the mechanical properties of co-extruded compound profiles consisting of EN AW-6082 and 20MnCr5 with previously published data [18].

Weidenmann et al. used the spring steel X10CrNi18-10 in the form of wires as reinforcement in co-extruded aluminum–steel compound profiles. The authors attributed the good bonding properties between the spring steel and the aluminum alloy EN AW-6060 to a metallurgical bond established via intermetallic phases whose formation is essentially influenced by the high contents of the alloying elements chromium and nickel [19]. Since this exact steel grade used was not available in the form of tubes with the intended dimensions, the reinforcing elements for the LACE experiments in the present study were made of the stainless steel X5CrNi18-10 with similar chromium and nickel contents. The chemical compositions of the materials used were determined by spark spectrometry (Table 1 and Table 2). The measurements showed that the actual composition of the stainless steel was well within the nominal values according to DIN EN 10088-1:2014-12. Moreover, it was confirmed that the contents of chromium and nickel, which were of particular interest with regard to metallurgical bonding, were similar to the alloy used by Weidenmann et al. [19].

A 10 MN extrusion press (SMS Meer GmbH, Düsseldorf, Germany) was used to perform the LACE experiments. An unheated modified tool holder was employed that allowed for a lateral feeding of reinforcing elements of various shapes. For the hollow LACE profiles produced in the present study, the effective extrusion ratio of the aluminum component was calculated by taking the inner diameter of the container (146 mm), the opening diameter of the die (62.68 mm), and the outer diameter of the steel tube (44.5 mm) into account. The initial diameter of the machined aluminum billet was 142 mm. During upsetting of the billet, the inner diameter of the container of 146 mm was reached so that the effective extrusion ratio of the current LACE process was 11:1. The steel tubes had an inner diameter of 32 mm, which corresponded to an overall reinforcement content of 34 vol. % for the compound profile. The steel tubes were ground with sandpaper and cleaned with ethanol prior to the LACE process. The conventional aluminum billets were preheated to a temperature of 530 °C for 4.5 h in a convection oven. The steel tubes were at room temperature at the beginning of the experiment. The LACE tool was preheated to a temperature of 490 °C, whereas the temperature of the container was 440 °C. In order to fill the tool as quickly as possible during the initial stage of each LACE experiment and ensure an almost isothermal process, a significantly higher ram speed of 1.5 mm/s was selected in the first phase of the process. During the filling of the tool with aluminum, the ram speed was gradually reduced to the intended test speed of 0.3 mm/s. This velocity was maintained for the rest of the LACE experiment. Compared with parameter sets typical for the extrusion of aluminum, the current LACE process is characterized by the combination of high temperature regimes and low ram velocities, both of which are based on insights gained from analogy tests in preliminary tests. In these experiments, a sufficient material bond between the joining partners was generally achieved when using high temperatures and long contact times [20].

### 2.2. Determination of the Mechanical Properties of the Interface

The mechanical properties of the interface between the joining partners of the produced compound profiles were examined using POTs as well as SCTs, for which samples were taken over the entire length of the compound profiles following the steps described in related work [18]. There, it has already been shown that there is a force and form fit between the two joining partners, aluminum and steel, in the hollow sections manufactured using LACE. However, prior to the current study there was only the indication of material bonding based on the failure of the sample inside the aluminum alloy due to macroscopic shear. Since the separation of the specimens in the SCT generally did not occur in the joint zone between aluminum and steel, the objective of the current study was to provide actual evidence of metallurgical bonding. Thus, the SCTs were also used to determine promising candidates for the subsequent detailed microstructural investigations using Scanning Electron Microscopy (SEM) and FIB based on the presence of failure inside the aluminum component.

Since the different coefficients of thermal expansion of the two metals resulted in shrinkage of the aluminum onto the steel tube during cooling of the compound profile [21], it is presumed that there is not only form fit but also force fit with the coaxial arrangement of the materials in the compound profiles. In order to determine the contribution of the different joining mechanisms on the mechanical properties individually, two types of tests were used. First, the bond strength was determined over the entire profile cross-section by means of POTs. Secondly, segments were wire eroded from slices of the specimens’ cross-sections to eliminate the influence of form or force fit. These segments were tested in the SCT. In order to be able to investigate the material behavior over the entire profile length, specimen slices for the two different tests were taken along the profile, alternating between the samples intended for the POT or the SCT (Figure 3a). The first fully “intact” specimen for these investigations was taken 25 mm after the longitudinal weld seams (LWS) appeared macroscopically closed for the hybrid profile with X5CrNi18-10 and after 35 mm for the profile with 100Cr6. This individual starting point is marked in the schematic sketch shown in Figure 3a. Further slices were taken with a thickness of 15 mm, taking into account the saw cut and the possibility of clamping. By means of machining, all slices were then turned to a plane-parallel height of 10 mm each.

For the POT, centering of the specimens in the testing machine was facilitated by attaching a step to the punch that was adapted to the inner diameter of the steel tube, as shown in Figure 3b. To allow the largest possible contact area, the hybrid specimens were arranged in the center on a steel ring to provide the best possible support for the aluminum during the testing. During the POT, the force-displacement curves were recorded. The de-bonding shear strength was determined from the maximum force, *F*_max_, using [19]:τmax=Fmaxπdh
where *d* is the outer diameter of the steel tube and *h* is the height of the specimen.

The slices of the compound profile intended for the SCT were divided into segments by wire eroding. For the experiments with 20MnCr5 and X5CrNi18-10, this procedure resulted in two specimen segments without LWS and one specimen segment that contained two LWS (compare Figure 4a), in order to detect whether the splitting of the aluminum alloy and rebonding inside the welding chamber of the tool had any influence on the bond formation. Specifically, the samples containing 100Cr6 steel were used to examine the influence of LWS in greater detail using the SCT. For this purpose, a modified specimen extraction shown in Figure 4b was used. Hence, these slices were divided into one specimen without LWS, one specimen with one LWS and one specimen with two LWS. With the aid of a laser microscope (type *VK 9700*, Keyence, Neu-Isenburg, Germany), the actual length of the interface present was measured and used for the calculation of the area of the joint zone. To enable uniaxial loading without tilting of the specimen during the SCT, the segments were clamped in the setup shown schematically in Figure 4c. The reinforcement was ejected with a universal testing machine (*type Z250*, Zwick, Ulm, Germany). For both the POT as well as the SCT, a preload of 50 N and a cross-head speed of 2 mm/min were used. The break-off criterion for these experiments was defined as a drop in force of 80%.

### 2.3. Microstructural Characterization of the Interface

A field-emitter SEM (*Auriga*, Carl Zeiss Microscopy GmbH, Oberkochen, Germany) equipped with an FIB was employed to prepare lamellae from the interface from one sample for each material combination. A gallium ion beam was used for the FIB preparation. In order to counteract the different removal rates of the aluminum and steel materials as best as possible, the specimens were aligned with the steel side facing the gallium ion beam. This was intended to reduce the faster removal rate of aluminum in order to achieve uniformly shaped lamellae. As shown in Figure 5 by the different transparency within the sample, the bottom side of the lamella had a thickness of 50 nm providing for very good lateral resolution during Energy-Dispersive X-ray Spectroscopy (EDS). The upper sides of the lamellae were cut to feature a thickness of 400 nm, and therefore EDS reduced lateral resolution but improved signal-to-noise ratio. The interface between the joining partners was imaged using both Backscattered Electrons (BSE) as well as Scanning Transmission Electron Microscopy (STEM).

The same samples that exhibited the highest shear strengths in the SCT were used for this particular analysis. This was based on the assumption that a sufficiently narrow phase seam significantly improves the bonding of the materials to each other, which, as a result, increases the strength of the compound. In order to achieve very good lateral resolution for the detection of the possibly very thin intermetallic phase seams, the side of the lamella that had a thickness of only 50 nm was used for STEM. Since the scattering processes of the electrons take place in a significantly reduced volume compared to bulk material, a higher lateral resolution of the elemental analysis can be achieved with the aid of EDS. The size of the diffusion zone created for each sample was determined from the slope of regression lines applied to measurement points within a ±0.1 μm distance around each point. The beginning of the diffusion zone was assumed to be at the point where the magnitude of the slope for aluminum and iron was continuously greater than 20 at. %·μm^−1^. Vice versa, the end of the diffusion zone was assumed to be where the amount of slope for aluminum and iron was lower than 20 at. %·μm^−1^.

## 3. Results

### 3.1. Evolution of the Extrusion Force during the LACE Process

For the LACE experiments, the evolution of the ram force over time was recorded and a typical progression is shown exemplarily for the material combination of EN AW-6082 and X5CrNi18-10 in Figure 6. As mentioned above, a higher ram speed was selected for initial filling of the tool compared to the actual extrusion process in order to counteract the cooling of the tool. Thus, a ram speed of ≈1.5 mm/s was used initially. The extrusion force increased rapidly and reached a first plateau at a force of 2.6 MN, which can be explained by the filling of the bridge part behind the inlets of the tool. At this stage of the process, the ram speed was reduced to ≈0.8 mm/s. Thereafter, the redirection of the aluminum in the tool took place and the force increased continuously, while the ram speed was reduced to 0.5 mm/s. The final target velocity of the actual aluminum extrusion was set to 0.3 mm/s when the extrusion force reached 5.0 MN, which caused a brief drop in force. Up to the point of the hybrid profile exiting the tool through the die, the force increased continuously up to 8.0 MN. The extrusion force remained relatively constant between 8.0 MN and 8.3 MN as the hybrid profile exited the die, until the end of the LACE experiment.

### 3.2. Mechanical Properties of the Compound Profiles

Using the POT and the SCT, the effective bonding mechanisms and the local mechanical properties of the compound profiles made of EN AW-6082 and either X5CrNi18-10 or 100Cr6 were determined. In Figure 7, the shear strengths resulting from both the POT and the SCT, which were carried out on samples according to Section 2.2, are shown in an alternating manner depending on the point of extraction. The results of the POT showed relatively consistent values over the entire profile length, starting from a maximum shear strength of 74 MPa at a distance of 25 mm to a value of 61 MPa at 235 mm. The minimum here was a value of 57 MPa, and an average strength of 63 MPa ± 5 MPa was observed in the POT. The shear strength determined by the SCT also showed similar values for most of the compound profile. The values at the beginning of the profile (40 mm of the aluminum cladding) were the only ones that differed significantly. Thus, the segments without LWS exhibited an average shear strength of 131 MPa and the segment with two LWS from the same profile section had a shear strength of 89 MPa. Moreover, these specimens had aluminum adhering to the reinforcing element over the entire sample circumference after the test, clearly indicating not only force and form closure, but also material bonding as an effective bonding mechanism. The average value for the shear strength at the end of the compound profile was nearly identical for segments both with (77 MPa) and without LWS (76 MPa). Apart from the outlier specimens taken at 40 mm and 130 mm after the beginning of the aluminum cladding, all different types of segments showed similar behavior during these tests. In general, the average shear strength of the current material combination, EN AW-6082 and X5CrNi18-10, was approximately 10 MPa larger overall compared with the compound profile consisting of EN AW-6082 and 20MnCr5, described in an earlier study [18].

The hybrid profile consisting of the material combination of EN AW-6082 and 100Cr6 was also tested with a POT and a SCT (Figure 8). In the POT, the variation of the determined shear strengths was more pronounced than was the case with the compound profile with X5CrNi18-10. The average strength of 44 MPa ± 8 MPa was lower than with the stainless steel. In contrast to the test series with X5CrNi18-10, the segments tested in the SCT exhibited either one LWS, two LWS or no LWS. The specimen with 47.5 mm behind the tip of the aluminum cladding was used to determine the exact positions of the LWS in this LACE profile and is therefore missing as a specimen for the SCT. The results of the SCT varied more compared to the POT. The specimens with one LWS generally had the lowest strengths over the entire profile length, with the minimum of 8 MPa at a distance of 172.5 mm. On average, the shear strength of these specimens with one LWS was 24 MPa ± 10 MPa. The segments without LWS varied the most, with an average shear strength of 36 MPa ± 15 MPa. The highest local shear strengths were obtained for this compound profile in a specimen with two LWS. Especially in the range from 247.5 mm to 347.5 mm of the aluminum cladding, the strengths were significantly higher and the maximum value of 69 MPa was determined at 347.5 mm. On average, the shear strength of the segments with two LWS was 50 MPa ± 12 MPa.

Table 3 summarizes the average mechanical properties of the hybrid profiles together with the specimens that showed highest shear strengths, which served as the basis for the subsequent detailed microstructural investigations. It is evident that the compound profiles with the different steel grades showed similar values in the POT when considering the standard deviation. Considering the standard deviations, it can only be stated that the compound profile with X5CrNi18-10 had higher shear strength than the one with 100Cr6. This tendency was also evident in the SCT. The higher-alloyed steel exhibited the highest shear strength of 131 MPa. The influence of the LWS on the bond formation could not be clarified, since the respective highest shear strengths resulted for specimens both with two LWS and with no LWS at all.

### 3.3. Characterization of the Interface

The targeted sample preparation using FIB lamellae of the joint zone was used to investigate the bonding mechanisms that led to the locally detected high shear strengths. It was assumed that a sufficiently narrow intermetallic phase seam in the joint zone resulted in better mechanical properties of the interface between aluminum and steel. For both the two material combinations examined in the current study and the material combination from a previous study [18], only a few sample segments exhibited outstanding shear strengths that were significantly above the average (Table 3). As shown in Figure 9, by way of an exemplary specimen consisting of EN AW-6082 and X5CrNi18-10 that featured a high strength, the two joining partners of aluminum and steel did not separate at the interface during the SCT. Instead, shearing was observed completely inside the aluminum component and a macroscopic aluminum layer was still firmly adhering to the steel.

For each material combination, one specimen segment with the highest strength was analyzed in detail using SEM. In particular, this included a specimen from the compound profile with 20MnCr5 reinforcement that had two LWS and was sampled at a distance of 100 mm. The mechanical properties of this material combination were already discussed in [18]. In case of the compound profile with X5CrNi18-10, a specimen segment without LWS taken 30 mm behind the tip of the aluminum cladding was used for the FIB preparation. For the FIB lamella of the material combination EN AW-6082/100Cr6 the segment with two LWS, taken at 347.5 mm of aluminum cladding, was used.

To allow for high-resolution element analysis, thin lamellae were prepared from the joint zones of these specimens using FIB. The STEM images of these lamellae are shown in Figure 10. A narrow phase seam was detected between the joining partners in all samples, which serves as proof for actual material locking as one of the acting bonding mechanisms at the investigated positions in these samples. In the case of the profile with EN AW-6082 and 20MnCr5, aggregated island-like structures can be observed in the joint zone (Figure 10a). These structures had a maximum width of 30 nm. By contrast, the joint zone between EN AW-6082 and the higher-alloyed steel X5CrNi18-10 showed a clearly pronounced and continuous phase seam with a width of up to 50 nm (Figure 10b). The sample with 100Cr6 also showed rather island-like structures in the joint zone (Figure 10c), which appeared less coagulated compared to the samples with either 20MnCr5 or X5CrNi18-10. However, this can be attributed to differences in the local thicknesses, etc., present in the individual lamellae, which represents only a small fraction of the joint zone of the sample. Here, the width was also about 30 nm.

The EDS analysis of the joint zone was performed at the significantly thicker location of the FIB lamella (ca. 400 nm) to obtain sufficient signal in a short measurement time, and thus minimize drift of the electron beam during the measurement. The change in the slope of the regression line was used to determine the diffusion range from aluminum to iron and vice versa. Based on a Monte Carlo calculation, a lateral EDS resolution of about 30 nm was assumed for both elements.

The EDS analysis, performed as a line scan, of the sample consisting of EN AW-6082 and 20MnCr5 is shown in Figure 11a. The corresponding BSE image does not show a clear transition from aluminum to iron. Instead, the joint zone exhibits different gray scales, which, in turn, indicate the diffusion of the elements into each other. The EDS analysis showed that aluminum continuously transitioned into the steel and vice versa. A plateau of the elements in the joint zone, indicating the formation of an intermetallic phase seam, could not be determined from this measurement. This is attributed to the lower lateral resolution when using thicker lamellae. Using the slope of the regression, a diffusion zone of the two elements into each other of about 1.0 µm was determined. No enrichments of other elements were detected in the joint zone. The EDS measurement of the EN AW-6082 and X5CrNi18-10 sample is displayed in Figure 11b. The BSE image shows a clear separation between aluminum and steel and the no transition zone is visible. In the EDS data, a continuous transition from aluminum to iron and vice versa could be determined. The diffusion zone determined via the slope of the regression line was approximately 0.9 µm and, hence, was similar to 20MnCr5. The alloying elements Cr and Ni had already diffused about 0.3 µm into the aluminum. An enrichment in the form of a peak in the joint zone was not detected. Figure 11c shows the measurement of the sample consisting of EN AW-6082 and 100Cr6. The STEM image showed a clear separation of the materials with no evidence of an intermetallic phase seam. Using EDS analysis, a continuous transition from aluminum to iron and vice versa was determined. The diffusion zone determined via the slope of the regression line here was approximately 0.6 µm, and thus was least pronounced.

In the case of the sample with 100Cr6, an EDS analysis was successfully conducted in an area where the lamella was only 40 nm thin, which led to good lateral resolution, without significant drift of the electron beam. This measurement (Figure 12) was performed within the diffusion range mentioned above. The higher lateral resolution already allowed the recording of an island-like structure in the interface between aluminum and steel in the STEM image. The EDS measurement revealed a small plateau of aluminum and iron, which indicates phase formation.

## 4. Discussion

Co-extruded semi-finished hybrid products made of EN AW-6082 and either X5CrNi18-10 or 100Cr6 have been investigated concerning their local mechanical properties as well as the effective bonding mechanisms. For comparison, a sample from a previous study [18], made of EN AW-6082 and 20MnCr5, which showed high strengths in the SCT, was analyzed by FIB and spatially resolved EDS analysis. The observed variations in the mechanical properties can be attributed mainly to a slightly uneven material flow of the aluminum alloy inside the tool resulting in different degrees of bonding. Since the examination of the specimen segments was assumed to be mostly independent from effects of form fit and force fit, the results of the SCT in particular were a clear indication that the highest strengths can be expected locally where material bonding through a diffusion zone or an already established but sufficiently narrow intermetallic phase seam is present.

The material combination EN AW-6082 and X5CrNi18-10 was selected based on the promising results that were reported in an earlier study [19]. The authors used the steel grade X10CrNi18-10, which has a similarly high content of the alloying elements chromium and nickel as the X5CrNi18-10 used in the present study. Compared with previous hybrid profiles made of EN AW-6082 and 20MnCr5, the current compression tests showed a slightly higher level of average shear strength for the hollow sections produced by LACE, with an average of 63 MPa ± 5 MPa over the entire profile length. The profile with 100Cr6 had an averaged strength of 44 MPa ± 8 MPa, which is similar to the value of 54 MPa ± 5 MPa that has been demonstrated previously for a compound profile consisting of EN AW-6082 and 20MnCr5 [18]. Hence, the shear strength of 61 MPa that was reported for wire-reinforced co-extruded profiles, determined by Weidenmann et al. by means of POT, could be reached by the as-extruded hybrid profiles in the present study. However, Weidenmann et al. achieved this value only by additional heat treatment after the co-extrusion. Although the LACE profiles should also have the potential for a further increase in strength by heat treatment, this has not yet been considered in this work, since premature strengthening of the hybrid semi-finished products would only be of limited use for the subsequent closed-die forging process [2].

Within the scope of this study, experimental proof of a firm bond between the joining partners of the co-extruded aluminum–steel compound was obtained. For this purpose, the SCT that eliminated the impact of force and form closure was carried out in order to obtain an indicator of material locking as the major bonding mechanism. Furthermore, samples that reached much higher stresses in the SCT compared with the more integral POT, were likely to have narrow intermetallic phase seams at their interfaces. This approach is based on the assumption of Herbst et al. that the strength of compounds consisting of aluminum and steel can be improved by narrow intermetallic phase seams < 1 µm [12]. While Weidenmann et al. proved the presence of intermetallic phases by means of transmission electron microscopy [22], no continuous intermetallic phase seams could be detected in the compound profiles shown there, even upon closer examination of the interface. Initially, intermetallic phases form in an island-like manner [11,13] and thus occur highly localized. Therefore, the present approach using samples that had highest strength was developed in order to detect the early stages of formation of the intermetallic phases via SEM. As it is evident from Figure 10a–c, the joining partners of all of the investigated material combinations have a narrow phase seam. In the case of the specimens with 100Cr6, this seam was still very localized, as indicated in Figure 1b. With 20MnCr5, the individual islands seem to have already grown together and the condition already corresponds to that between Figure 1b and 1c. In the case of the higher-alloyed X5CrNi18-10, a continuous phase seam could be detected, as indicated in Figure 1c.

By contrast, a pronounced plateau in the element distributions could not be detected with the relatively thick FIB lamellae used, since here the possibility of higher measurement times was in favor of signal-to-noise ratio but at the expense of lateral EDS resolution. For all material combinations, a gradual transition of the elements aluminum and iron into each other was observed. The determined thickness of the diffusion zone was similar for samples containing 20MnCr5 and X5CrNi18-10. In the case of specimens with the bearing steel 100Cr6, the diffusion zone was ≈ 0.3 µm smaller. Hence, the expected influence of the alloying elements on the bond formation [19] was not observed for the materials used in the present study. For the sample with 100Cr6 it was possible to perform an EDS analysis using the thinner FIB lamella, and thus to obtain an improved lateral resolution. In this case it could be shown that there was not only a continuous diffusion between aluminum and iron, but that there was already a very narrow plateau due to an intermetallic phase seam. The exact phase could not be determined, but the formation of FeAl_3_ or Fe_2_Al_5_ was expected [10] as the stoichiometry of the EDS indicated the formation of these phases [23]. For the combination of a 6xxx aluminum alloy with Fe, the formation of β-AlFeSi or α-AlFeSi, with β-AlFeSi usually transforming to α-AlFeSi, was assumed [24], but could not be observed. It could be shown that the proportion of silicon already increased in the phase seam, but then remained at one level. An enrichment indicating an Al_x_Fe_y_Si_z_ phase could not be detected. Therefore, a saturation by diffusion was assumed, as expected for aluminum alloys with a high silicon content [23].

Weidenmann et al. attributed the formation of the compound mainly to the elements chromium and nickel [19]. The 20MnCr5 and 100Cr6 investigated here did not contain nickel, which is why the formation of the compound cannot be attributed to this. The content of chromium in 20MnCr5 and 100Cr6 is much lower than in X5CrNi18-10. Since the EDS analysis did not show any enrichment of chromium in the interface, the assumption of the high influence of this alloying element on the bond formation could not be verified. Moreover, the accumulation of silicon in the interface shown by Liu et al. [25] and Wang et al. [26] for friction welded samples of aluminum and steel was not observed. The formation of intermetallic phases of the Al_x_Fe_y_Si_z_ type with silicon intercalation [27] was not observed for the co-extruded samples.

For the use of the semi-finished products in the *Tailored Forming* process chain, the very narrow phase seam or the beginning of the formation of island-like intermetallic phases in the interface that has been demonstrated here is advantageous. It can be assumed that subsequent steps, such as a die forging and heat treatment, can improve this connection without the phase seam in the interface having a negative influence on the properties of the hybrid components. In order to provide clear evidence of the emerging phases in the future, measurements should be carried out using electron backscattered diffraction on the FIB lamellae. In this way, the lattice structure can be used to prove whether FeAl_3_ or Fe_2_Al_5_ are formed, and whether the parameters are changed by the incorporation of silicon.

## 5. Conclusions

A Lateral Angular Co-Extrusion process for the production of hybrid hollow sections was successfully transferred to new material combinations of the wrought alloy EN AW-6082 paired with either the higher-alloyed steel X5CrNi18-10 or the bearing steel 100Cr6. The investigations were intended to provide evidence of material bonding by materials joined exclusively by co-extrusion with no further heat treatments. The joint zone of a compound profile with 20MnCr5, of which the mechanical properties were investigated in a previous study [18], was used as a reference. The main results can be summarized as follows:For the material combination EN AW-6082/X5CrNi18-10, a high strength of up to 131 MPa could be demonstrated in SCT on segments cut from the compound profile;Using FIB lamellae, it was possible to locally investigate the interface of shear compression specimens that exhibited particularly high strength. It could be demonstrated that there was a metallurgical bond between the aluminum alloy EN AW-6082 and the different steel grades employed;While 20MnCr5 and X5CrNi18-10 showed a diffusion zone as well as local intermetallic phases in the form of narrow island-like structures, the higher-alloyed steel X5CrNi18-10 showed a continuous phase seam as well as a broad diffusion zone.An enrichment of the main alloying elements of the steel (chromium and nickel) near the interface could not be confirmed. However, the bond appeared to be microstructurally more homogenous with X5CrNi18-10 than with 20MnCr5 and 100Cr6.

## Figures and Tables

**Figure 1 materials-15-01692-f001:**
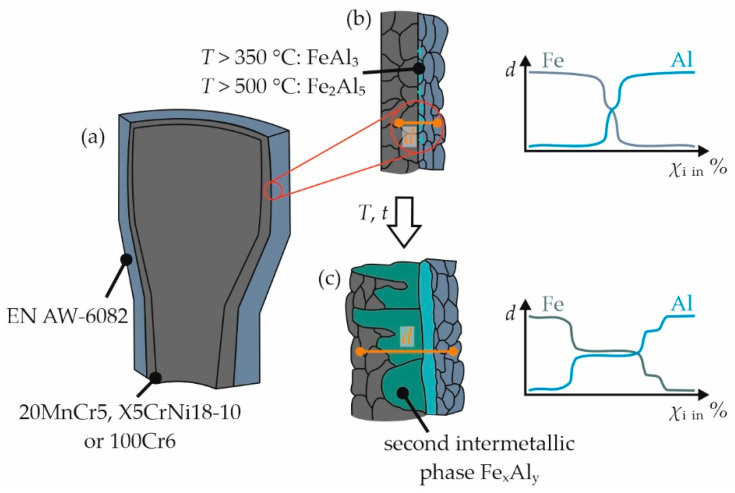
Schematic illustration of (**a**) the interface between aluminum and steel of a hybrid bearing bushing and (**b**) exemplary course of the mass fraction *χ*_i_ of aluminum and iron along a measuring section *d* through the different phases at the beginning and (**c**) after prolonged exposure to elevated temperature; see main text for details.

**Figure 2 materials-15-01692-f002:**
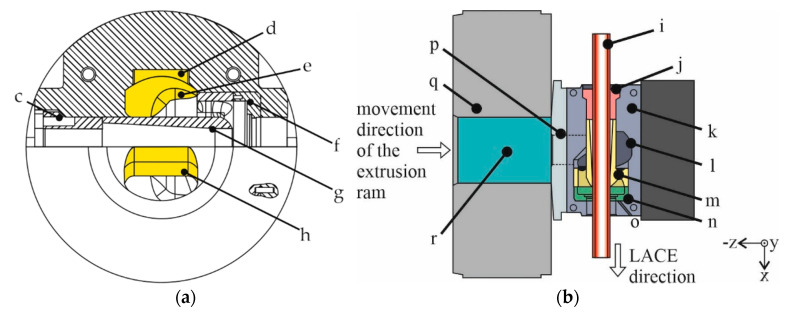
Scheme of a (**a**) sectional plane lengthwise through the LACE tool for a detailed visualization of the components, with c: clamping cover, d: pocket, e: deflection, f: die, g: mandrel part with three support arms, h: entry (the geometry of the entry and deflection are highlighted); (**b**) longitudinal section of the experimental setup for the 10 MN extrusion press with i: reinforcing element, j: clamping cover, k: tool housing, l: deflection, m: mandrel part with three support arms, n: die, o: thermocouple bore, p: bridge, q: container, r: aluminum billet; for additional details see [2,18].

**Figure 3 materials-15-01692-f003:**
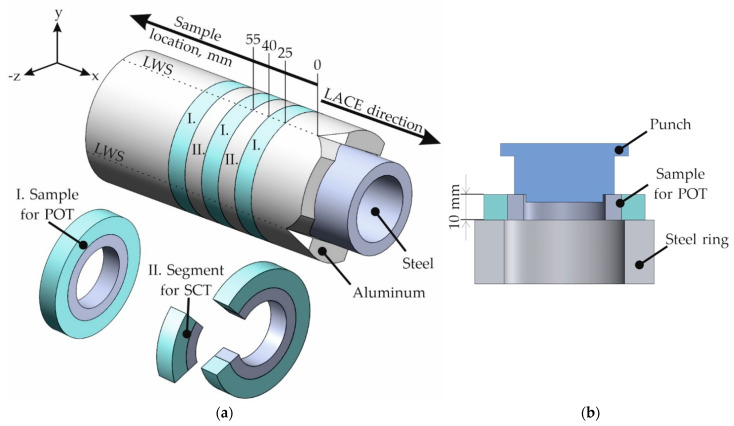
Schematic illustration of (**a**) a compound profile with alternating sampling for the characterization of the mechanical properties using push-out tests (POTs) and shear compression tests (SCTs); (**b**) setup for the push-out tests (POTs).

**Figure 4 materials-15-01692-f004:**
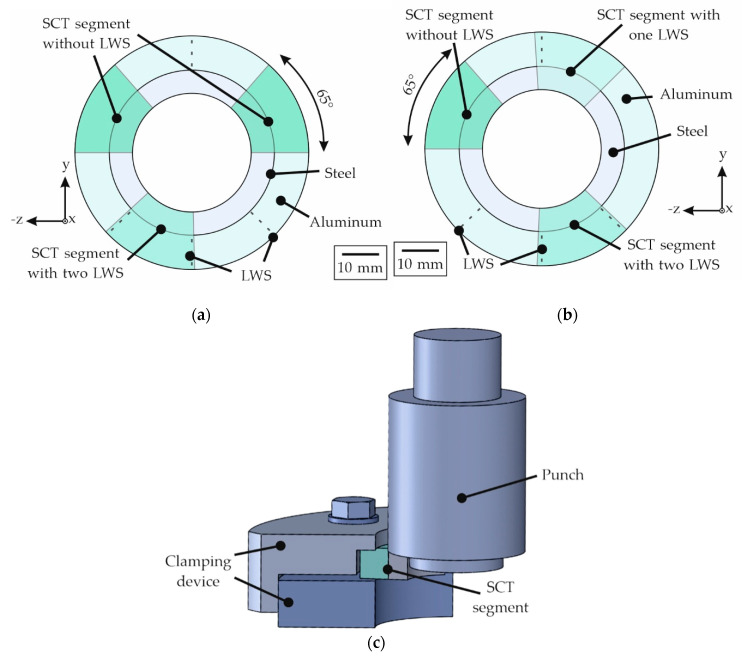
Schematic illustration of: (**a**) a compound sample from 20MnCr5 and X5CrNi18-10 with one segment with two LWS and two segments without LWS (highlighted by dashed lines); (**b**) a compound sample with 100Cr6 with one segment with two LWS, one segment with one LWS and one segment without LWS (highlighted by dashed lines); (**c**) the experimental setup for the shear compression test (SCT).

**Figure 5 materials-15-01692-f005:**
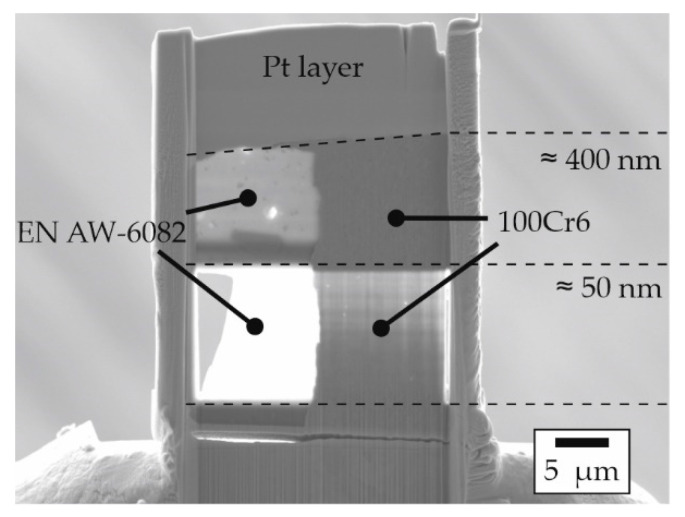
Example of a lamella with a thickness of approx. 400 nm at the top and 50 nm at the bottom; here with the material combination EN AW-6082 and 100Cr6.

**Figure 6 materials-15-01692-f006:**
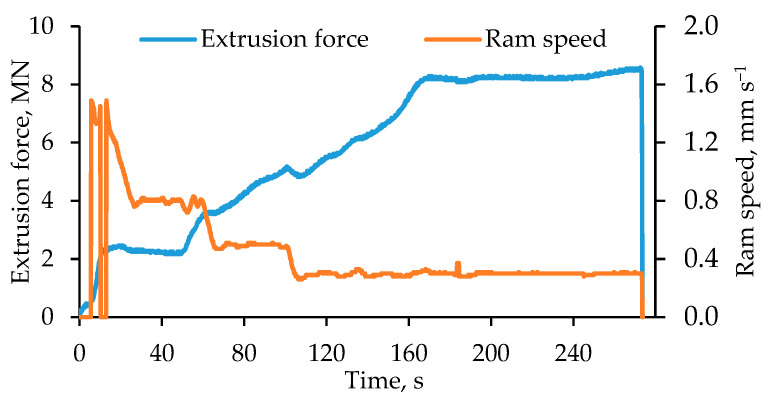
Evolution of the extrusion force during the duration of a LACE experiment with an extrusion ratio of 11:1, using EN AW-6082 and X5CrNi18-10.

**Figure 7 materials-15-01692-f007:**
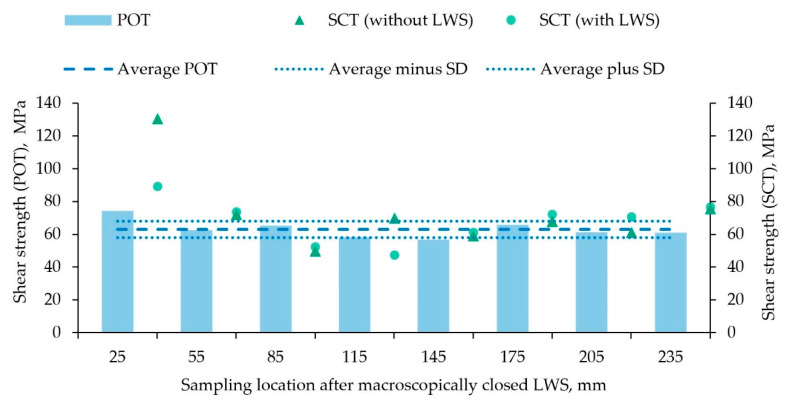
Shear strengths along the compound profile made of EN AW-6082 and X5CrNi18-10 determined on alternating samples tested in push-out tests (bar chart; POTs) or in shear compression tests (SCTs) of segments with (dots) and without (triangles) LWS; sampling began 25 mm behind the point, where the LWS appeared macroscopically closed; the dashed line shows the shear strength averaged over the entire profile length as measured in the POT; the dotted lines are the upper and lower limits of the corresponding standard deviation (SD).

**Figure 8 materials-15-01692-f008:**
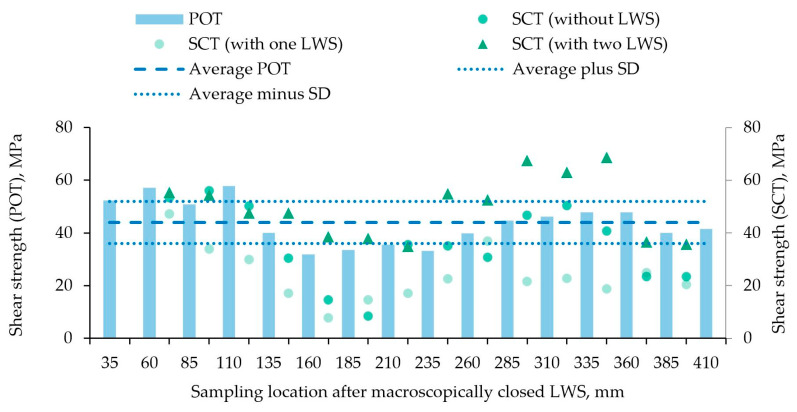
Shear strengths along the compound profile made of EN AW-6082 and 100Cr6 determined on alternating samples tested in push-out tests (bar chart; POTs) or in shear compression tests (SCTs) of segments with (dots) and without (triangles) LWS; sampling began 35 mm behind the point, where the LWS appeared macroscopically closed; the dashed line shows the shear strength averaged over the entire profile length as measured in the POT; the dotted lines are the upper and lower limits of the corresponding standard deviation (SD).

**Figure 9 materials-15-01692-f009:**
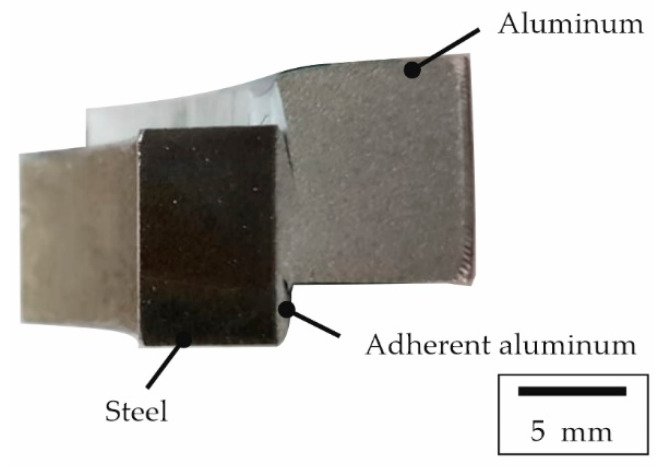
Sample of the material combination EN AW-6082 and X5CrNi18-10 after the SCT with failure inside the aluminum alloy next to the joint zone and aluminum adhering on the reinforcement.

**Figure 10 materials-15-01692-f010:**
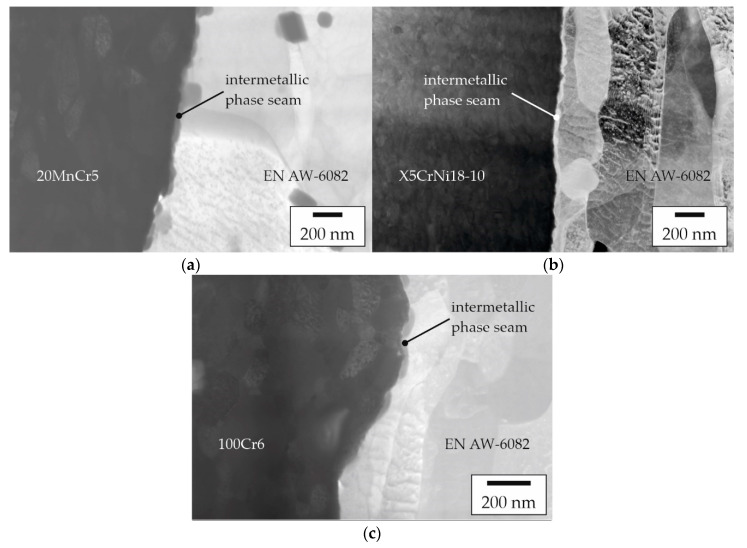
STEM images of the joint zone between the joining partners: (**a**) EN AW-6082 and 20MnCr5, (**b**) EN AW-6082 and X5CrNi18-10 with intermetallic phase seams, and (**c**) EN AW-6082 and 100Cr6 with incipient island-like growth of intermetallic phases.

**Figure 11 materials-15-01692-f011:**
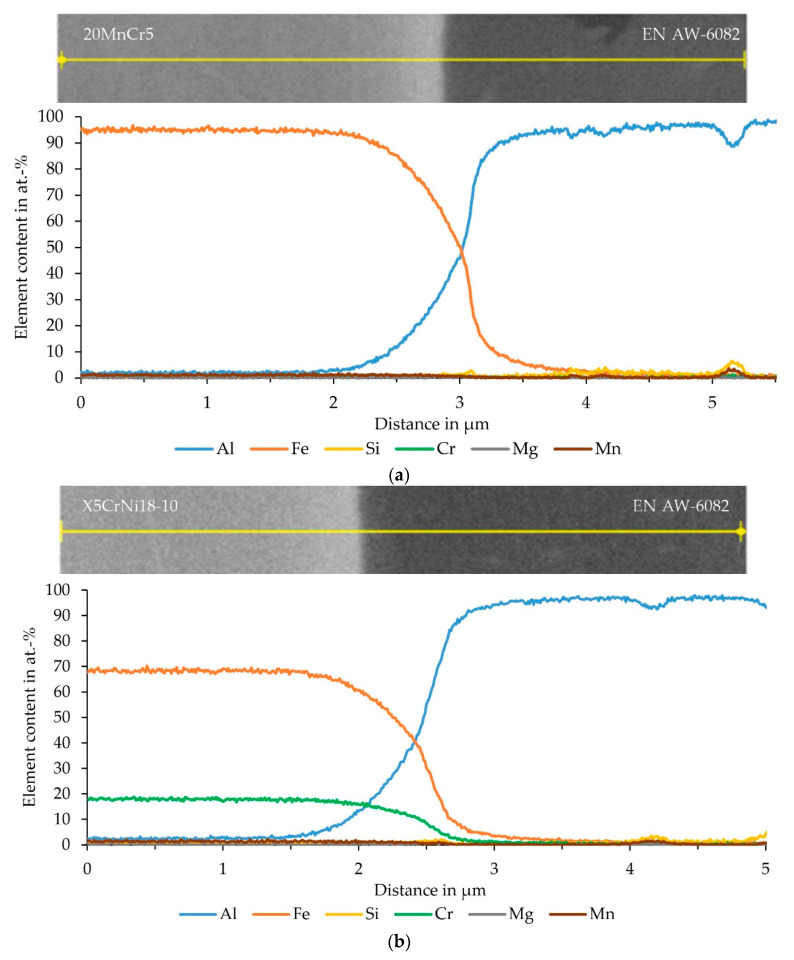
BSE images with EDS measuring range and determined element distribution of a thick (≈ 400 nm) FIB lamella, prepared from the interface of the specimen segment for SCT, which exhibited particularly high strength, consisting of (**a**) EN AW-6082 and 20MnCr5, (**b**) EN AW-6082 and X5CrNi18-10 and (**c**) EN AW-6082 and 100Cr6 (in this case a STEM image is shown).

**Figure 12 materials-15-01692-f012:**
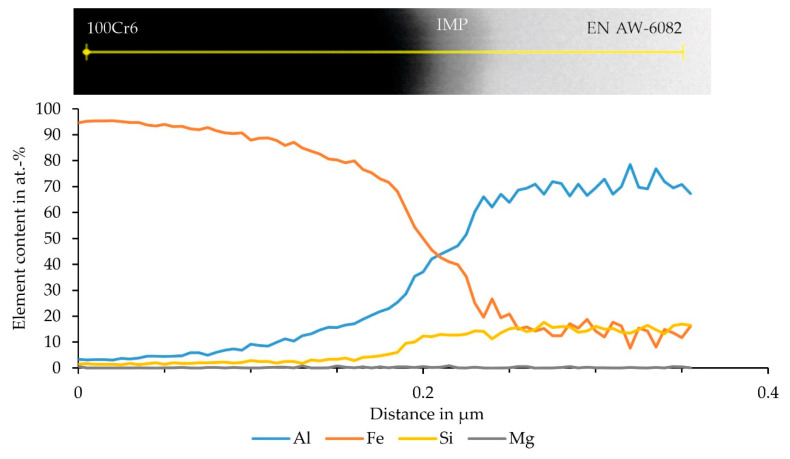
STEM image with EDS measuring range and determined element distribution of the thin FIB lamella, prepared from the interface of the specimen segment for SCT, which exhibited particularly high strength, consisting of EN AW-6082 and 100Cr6 with an island-type intermetallic phase (IMP) in the middle.

**Table 1 materials-15-01692-t001:** Chemical composition in wt. % of the EN AW-6082 billet used, balance Al.

	Si	Fe	Cu	Mn	Mg	Cr	Zn	Ti
Sample EN AW-6082	1.30 ± 0.04	0.20 ± 0.007	0.08 ± 0.02	0.76 ± 0.02	0.86 ± 0.08	0.02 ± 0.001	0.05 ± 0.001	0.02 ± 0.001

**Table 2 materials-15-01692-t002:** Chemical compositions in wt. % of the steel grades 20MnCr5, X5CrNi18-10, and 100Cr6 used for the LACE experiments, compared to the compositions specified in the standards DIN EN 10088-1:2014-12 and DIN EN ISO 683-17.

	C	Si	Mn	P	S	Cr	Mo	Ni	N
Sample 20MnCr5	0.209 ± 0.01	0.245 ± 0.01	1.19 ± 0.01	0.011 ± 0.01	0.027 ± 0.01	1.11 ± 0.01	-	-	-
Sample X5CrNi18-10	0.032 ± 0.01	0.392 ± 0.01	1.47 ± 0.01	0.025 ± 0.01	0.023 ± 0.01	18.61 ± 0.05	-	8.95 ± 0.09	0.07 ± 0.01
Sample 100Cr6	0.971 ± 0.02	0.234 ± 0.01	0.334 ± 0.01	0.012 ± 0.01	0.008 ± 0.01	1.48 ± 0.02	0.018 ± 0.01	0.075 ± 0.01	-
Standard X5CrNi18-10	0.07	1.00	2.00	0.045	0.03	17.00–19.50	-	8.00–10.50	0.10
Standard X10CrNi18-10	0.05–0.15	2.00	2.00	0.045	0.03	16.00–19.00	0.8	6.00–9.50	0.10
Standard 100Cr6	0.93–1.05	0.15–0.35	0.25–0.45	0.025	0.015	1.35–1.60	0.1	0.3	-

**Table 3 materials-15-01692-t003:** Average shear strength from the POT and max. shear strength determined in SCT of sample segments.

Material Combination	Average Shear Strength (POT), MPa	Max. Shear Strength (SCT) of the Sample Used for Further Investigations, MPa
EN AW-6082/20MnCr5 [18]	54 ± 5	92
EN AW-6082/X5CrNi18-10	63 ± 5	131
EN AW-6082/100Cr6	44 ± 8	69

## Data Availability

Data are available upon request from the corresponding author.

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
