# Peer review of "Characterization of the Interface between Aluminum and Iron in Co-Extruded Semi-Finished Products"

_materials, 2022, doi:10.3390/ma15051692_

Round 1
Reviewer 1 Report
Characterization of the interface between aluminum and iron in co-extruded semi-finished products was the aim of the article. A new material combinations of the EN 558 AW-6082 alloy paired with either the higher alloyed steel X5CrNi18-10 or the bearing steel 100Cr6 were checked. The issues discussed in the article are undoubtedly of interest in both scientific as well as technological point of view. The composition of the article meets the journal's recommendations and includes the following sections: Abstract, Introduction, Materials and methods, Results, Discussion, Conclusions and References. Generally, the applied research techniques and posted results confirm the presence of the intermetallic phases between materials joined by co-extrusion with no further heat treatments.
In the opinion of the reviewer, only the minor revision before publication is required:
- in some parts of the text there is no reference to the cited publications (eg. line 43: BACH et al. have already carried out research on the production…, line 161: WEIDENMANN et al. used the spring steel…),
- I propose to add an abbreviation for selected terms, mentioned many times in the text [eg. line 128: push-out tests (POT), line 130: shear compression tests (SCT)] - and use the shortcut later,
- The number of section “3.2. Characterization of the interface” should be updated to 3.3.
Author Response
Dear editor,
Thank you for sending the reports from the reviewers of our manuscript “Characterization of the interface between aluminum and iron in co-extruded semi-finished products”.
We appreciate the opportunity to revise the manuscript. The comments and suggestions were very helpful to us and we thank the reviewers for their careful review of the paper. In the revised version of the manuscript, we have taken all comments into account as described below. All changes in the revised manuscript were made using the "track changes" mode. The changes made at the request of the reviewers are additionally marked in yellow. The list of references has been expanded as suggested by the reviewer 2. The new references are also marked in the list of references and the reference numbers in the text have been updated. For sake of clarity, the later have not been marked in yellow.
Response Reviewer 1:
Point 1: In some parts of the text, there is no reference to the cited publications (eg. line 43: BACH et al. have already carried out research on the production…, line 161: WEIDENMANN et al. used the spring steel…)
Response 1: We are not sure what went wrong here. In our original file, both references were already addressed in the paper, and we have highlighted the corresponding text on pages 2 and 4 now. We would certainly double-check this again in the final proofs.
Point 2: I propose to add an abbreviation for selected terms, mentioned many times in the text [eg. line 128: push-out tests (POT), line 130: shear compression tests (SCT)] - and use the shortcut later.
Response 2: The abbreviations POT and SCT are now defined and used as suggested.
Point 3: The number of section “3.2. Characterization of the interface” should be updated to 3.3.
Response 3: Thanks for pointing this out. We have now corrected the enumeration.

Reviewer 2 Report
To the Materials
Dear Ms. Cathy Zeng,
Thank you for inviting me for the review of the manuscript “Characterization of the interface between aluminum and iron in co-extruded semi-finished products”.
General review: This author explains the hybrid hollow profiles, made of the aluminum alloy EN 16 AW-6082 paired with either the case-hardening steel 20MnCr5, the stainless steel X5CrNi18-10, or the rolling bearing steel 100Cr6, were produced by Lateral Angular Co-Extrusion. Push-out tests on hybrid hollow sections over the entire sample cross-section have shown shear strengths of 44 MPa 19 ± 8 MPa (100Cr6) up to 63 MPa ± 5 MPa (X5CrNi18-10). In particular, the influence of force and form closure on the joint zone could be determined using specimen segments tested in shear compression. Locally, shear strengths of up to 131 MPa (X5CrNi18-10) were demonstrated in the shear compression test. From these samples, lamellae for microstructural analysis were prepared with a Focused Ion Beam. Detailed analyses showed that for all material combinations a material bond in the form of an ultra-thin intermetallic phase seam with a thickness of up to 50 nm could be established.
Minor revision:
- For the shear test, can you mention the ASTM regulation number or any basis knowledges because the experiment for the shear test seemed to be unique?
- So, what is the intermetallic phase with a thickness of up to 50 nm? If they are expected to be FeAl3 or Fe2Al5, not AlxFeySiz, what is the reason although there are the other compounds? Do nickel or chromium in steel have influenced to those?
- For the welding of the two dissimilar materials, I strongly recommend to read and cite the manuscript entitled “Selective compositional range exclusion via directed energy deposition to produce a defect-free Inconel 718/SS 316L functionally graded material”, and to change from the very old references to the new ones as possible.
Otherwise, the author has addressed well most of the explanations. Overall, this is a good study. I supposed it can be considered for publication in Materials after minor revision.
Author Response
Dear editor,
Thank you for sending the reports from the reviewers of our manuscript “Characterization of the interface between aluminum and iron in co-extruded semi-finished products”.
We appreciate the opportunity to revise the manuscript. The comments and suggestions were very helpful to us and we thank the reviewers for their careful review of the paper. In the revised version of the manuscript, we have taken all comments into account as described below. All changes in the revised manuscript were made using the "track changes" mode. The changes made at the request of the reviewers are additionally marked in yellow. The list of references has been expanded as suggested by the reviewer 2. The new references are also marked in the list of references and the reference numbers in the text have been updated. For sake of clarity, the later have not been marked in yellow.
Response Reviewer 2:
Point 1: For the shear test, can you mention the ASTM regulation number or any basis knowledges because the experiment for the shear test seemed to be unique?
Response 1: The push-out test were based on earlier work by Weidenmann et al., who performed such tests on extruded specimens with a wire reinforcement. Based on these investigations, which we transferred to the specimens with flexurally rigid steel tubes, we developed the shear compression tests to determine local phenomena. Despite intensive literature research, we could not find a suitable ASTM standard for these specific tests.
Point 2: So, what is the intermetallic phase with a thickness of up to 50 nm? If they are expected to be FeAl3 or Fe2Al5, not AlxFeySiz, what is the reason although there are the other compounds? Do nickel or chromium in steel have influenced to those?
Response 2: Thank you for pointing out this important aspect. The other elements are the components of the aluminium alloy used or the steel examined in each case. No enrichments of alloying elements such as Cr or Mn could be detected (compare page 18, lines 533-538). The following text was amended in the discussion (page 18, lines 519-532):
„Hence, the expected influence of the alloying elements on the bond formation [19] was not observed for the materials used in the present study. For the sample with 100Cr6 it was possible to perform an EDS analysis using the thinner FIB lamella, and thus to obtain an improved lateral resolution. In this case it could be shown that there is not only a continuous diffusion between aluminum and iron, but that there is already a very narrow plateau due to an intermetallic phase seam. The exact phase could not be determined, but the formation of FeAl3 or Fe2Al5 is expected [10] as the stoichiometry of the EDS indicate the formation of these phases [23]. For the combination of a 6xxx aluminum alloy with Fe, the formation of β-AlFeSi or α-AlFeSi, with β-AlFeSi usually transforming to α-AlFeSi, can be assumed [24], but could not be observed. It could be shown that the proportion of silicon already increases in the phase seam, but then remains at one level. An enrichment indicating an AlxFeySiz phase could not be detected. Therefore a saturation by diffusion is assumed, as expected for aluminum alloys with a high silicon content [23].“
Furthermore, a sentence was added to the end of the discussion section for possible further EBSD investigations, which could not yet be carried out in this study. In this way, the question of which phase is actually present in the intermetallic phase seam could be clarified in the future.
Point 3: For the welding of the two dissimilar materials, I strongly recommend to read and cite the manuscript entitled “Selective compositional range exclusion via directed energy deposition to produce a defect-free Inconel 718/SS 316L functionally graded material”, and to change from the very old references to the new ones as possible.
Response 3: Thank you very much for the advice. We have included this and additional work on the topic to better compare our work with other recent works. We have still kept the earlier works, such as Ryabov, Achar or Eggeler, as these are the primary sources.
